# Might Fibroblasts from Patients with Alzheimer’s Disease Reflect the Brain Pathology? A Focus on the Increased Phosphorylation of Amyloid Precursor Protein Tyr_682_ Residue

**DOI:** 10.3390/brainsci11010103

**Published:** 2021-01-14

**Authors:** Filomena Iannuzzi, Vincenza Frisardi, Lucio Annunziato, Carmela Matrone

**Affiliations:** 1Department of Biomedicine, University of Aarhus, Bartholins Allé, 8000 Aarhus, Denmark; Filomena.iannuzzi@biomed.au.dk; 2Geriatric and Neuro Rehabilitation Department, Clinical Center for Nutrition in the Elderly, AUSL-IRCCS Reggio Emilia, Giovanni Amendola Street, 42122 Reggio Emilia, Italy; Vincenza.Frisardi@ausl.re.it; 3SDN Research Institute Diagnostics and Nuclear (IRCCS SDN), Gianturco, 80131 Naples, Italy; lannunzi@unina.it; 4Division of Pharmacology, Department of Neuroscience, School of Medicine, University of Naples Federico II, 80131 Naples, Italy

**Keywords:** Alzheimer’s disease, amyloid precursor protein, Tyr682 residue, YENPTY motif, Fyn tyrosine kinase, amyloid beta

## Abstract

Alzheimer’s disease (AD) is a devastating neurodegenerative disorder with no cure and no effective diagnostic criteria. The greatest challenge in effectively treating AD is identifying biomarkers specific for each patient when neurodegenerative processes have not yet begun, an outcome that would allow the design of a personalised therapeutic approach for each patient and the monitoring of the therapeutic response during the treatment. We found that the excessive phosphorylation of the amyloid precursor protein (APP) Tyr_682_ residue on the APP _682_YENPTY_687_ motif precedes amyloid β accumulation and leads to neuronal degeneration in AD neurons. We proved that Fyn tyrosine kinase elicits APP phosphorylation on Tyr_682_ residue, and we reported increased levels of APP Tyr_682_ and Fyn overactivation in AD neurons. Here, we want to contemplate the possibility of using fibroblasts as tools to assess APP Tyr_682_ phosphorylation in AD patients, thus making the changes in APP Tyr_682_ phosphorylation levels a potential diagnostic strategy to detect early pathological alterations present in the peripheral cells of AD patients’ AD brains.

## 1. Introduction

Alzheimer’s disease (AD) is a prominent neurodegenerative disease which takes a major toll on the elderly and places an enormous burden on the healthcare system.

The amyloid precursor protein (APP) is one of most extensively studied molecules in AD, and its cleavage, mediated by β-site APP-cleaving enzyme 1 (BACE1 or β-secretase) and by γ-secretase, leads to the dysregulated production of Aβ peptides, which mostly accumulate at central synapses [1].

Our group have previously underlined the crucial role of APP Tyr_682_ residue in the processes responsible for the generation of Aβ peptides in human neurons [2,3]. APP Tyr_682_ residue is located in the highly conserved _682_YENPTY_687_ motif, which binds specific adaptor proteins depending on its phosphorylation/dephosphorylation state [4]. The APP interaction with these proteins acts as the major regulator of APP fate [5,6,7]. In particular, increased APP Tyr_682_ phosphorylation prevents the APP from binding to the Clathrin and AP2 endocytic proteins [8,9], and affects APP endocytosis and trafficking inside neurons [8,9]. Consequently, APP accumulates in the acidic neuronal compartments, such as late endosomes and lysosomes, where it is preferentially cleaved to generate Aβ peptides [3]. Notably, we previously showed that high levels of phosphorylation of APP Tyr_682_ residue in neurons differentiated from neuronal stem cells of AD patients [2,3,10] preceding Aβ accumulation, and promoting neuronal degeneration. We therefore suggested that this high levels of APP Tyr_682_ phosphorylation might be an early marker of neuronal degeneration [5,6]. Furthermore, we identified Fyn tyrosine kinase (TK) in AD neurons as being responsible for the increased levels of APP Tyr_682_ phosphorylation, that in turn triggers biochemical events downstream, finally resulting in Aβ production and neuronal death [2]. Consistently, the reduction in Fyn activity, using Fyn TK inhibitors (TKI) or Fyn mRNA silencing, led to the decrease in APP Tyr_682_ phosphorylation and Aβ release in AD neurons [2,3]. These findings opened up a promising perspective for the use of APP Tyr_682_ residue as a biomarker to identify patients with high levels of APP Tyr_682_ phosphorylation that may benefit from personalised treatments with Fyn TKI.

We found elevated APP Tyr phosphorylation levels and Fyn overactivation both in skin fibroblasts and cortical tissues of minipigs carrying PSEN1 M146I mutation [3]. The increased APP Tyr phosphorylation in fibroblasts of mutant minipigs was associated to the same defects in APP trafficking that we also reported in neurons from AD patients carrying the same PSEN1 M146L mutation [3]. Relevantly, both APP Tyr phosphorylation and the defects in APP trafficking were rescued by Fyn TKI [2,3]. Of note, minipigs carrying PSEN1 M146L mutation displayed Aβ plaques and tau tangles in brain [11]. This evidence suggested the existence of possible common molecular pathways between fibroblasts and AD neurons, and encouraged us to explore the possibility of using skin fibroblasts to assess the levels of APP Tyr phosphorylation in AD patients. Indeed, the need to identify changes in peripheral cells recapitulating genetic, epigenetic, or metabolic deficiencies recognised in the brains of AD patients, and to develop standardised procedures to analyse changes in these cellular populations, is crucial for biomarker discovery.

## 2. Materials and Methods

### 2.1. Human Fibroblasts

Human fibroblasts were purchased from the Coriell Institute (New Orleans, LA, USA) and cultured following the guidelines reported on the manufacturer’s web site (https://catalog.coriell.org/). All fibroblast cultures derived from an ante mortem skin biopsy from the forearm, and had approximately the same biological age in culture (number of passages of cell lines and cell divisions). Fibroblast cell cultures grew very slow, but they all showed the same morphology and shape in culture. No differences were found in the extent of neuronal survival or proliferation. In addition, Aβ40 and Aβ42 were undetectable in media of fibroblast cultures, because the levels appeared below the sensitivity of a commercially available ELISA kit (data not shown).

Patients were classified according to the criteria reported on the Coriell website and are summarised in Table 1: AD, sporadic AD patients; fAD 1, patients carrying APP mutations or APP duplication on chromosome 21 [12]; and PSEN1, patients carrying Ala246Glu (A246E) PSEN1 mutation. As controls, based on age and sex of the donors, we included six samples from unaffected spouse and four from donors with a familiar history of AD (fAD 1) or one sibling (#AG06846) of a patient carrying the PSEN 1 mutation A246E (#AG06848). As further controls of our experiments, we included three patients with a clinical diagnosis of Parkinson’s disease (PD) (#442, #446, #527), and one of epilepsy (#159). In particular, donor #527 (PD + AD) was initially diagnosed as PD because he exhibited difficulties in movements. However, one year after diagnosis he developed problems with attention span and memory. Autopsy confirmed AD. All clinically diagnosed AD and PD patients—according to clinical or neuroimaging features—received histopathological confirmation at autopsy.

Some more information about patient clinical features and specific characteristics of the fibroblast cultures are available on the Coriell online catalogue (https://catalog.coriell.org/), using the source number (#) reported in Table 1.

### 2.2. Immunoprecipitation and Western Blot Assays

Fibroblasts were kept in culture until cell confluence was reached. Lysates were firstly processed for IP (pTyr) using mouse anti-p-Tyr magnetic beads, and subsequently analysed by WB. For pTyr protein enrichment, we used Phospho-Tyrosine Mouse mAb Magnetic Bead Conjugate (P-Tyr-100) (Cell Signaling, #8095). The samples were loaded on a 4–20% precast gel. Western blot analysis was performed using anti APP, clone Y188 (abcam, ab32136), rabbit anti-pan Fyn (Cell Signaling, BioNordika, Herlev, Denmark, #4023), rabbit anti-Src pTyr_416_ (Cell Signaling Technology, #2101), mouse anti-p-Tyr (Sigma-Aldrich, #9416) and monoclonal anti-β-actin-peroxidase (Sigma-Aldrich, A3854, Søborg, Denmark) antibodies.

### 2.3. Statistical Analysis

Data are expressed as mean ± standard error of the means. All the experiments were performed at least three times. The appropriate statistical test was selected using GraphPad Prism software (version 9.0c) (GraphPad, San Diego, CA, USA); details are reported in the legend for each figure.

## 3. Results

In the study, we included fibroblasts from ante mortem skin biopsies of 14 AD patients, six unaffected spouses, and four healthy donors with a familiar history of AD. In addition, four patients with a diagnosis of dementia, three associated to Parkinson’s disease (PD) and one to epilepsy were included. Post-mortem histopathology brain examinations were performed in all patients with mild cognitive impairment or dementia diagnosis to confirm AD.

The levels of phosphorylation of APP Tyr were assessed using the same procedure previously applied to neurons [2,3] and consisting of immunoprecipitating the total lysates with mouse anti pTyr conjugated beads and analysing samples by Western blot (WB) using anti APP antibody. Although this procedure is not selective for APP Tyr_682_ residue (APP has indeed other two Tyr(s) whose changes in the phosphorylation can also easily be detected by the same procedure), previous findings from our group proved that only APP Tyr_682_ residue phosphorylation is crucial in initiating APP amyloidogenic processes in AD neurons, whereas Tyr phosphorylation at different APP sites does not affect the extent of A𝛽 production or accumulation [8,9]. The increase in APP Tyr phosphorylation was analysed with respect to the basal APP expression levels in the total lysate of each sample (Figure 1, Figure 2, Figure 3 and Figure 4). In addition, the Fyn Tyr_420_ phosphorylation was also assessed as an indirect measure of Fyn TK activity [13], and the results were normalised on Fyn basal expression levels (Figure 1, Figure 2, Figure 3 and Figure 4).

We found a significant increase in the level of APP Tyr phosphorylation in the majority of fibroblasts from AD patients when compared to healthy donors, as well as patients with other dementias (Figure 1). In particular, APP Tyr phosphorylation levels were significantly increased in all AD patients when compared to healthy unaffected spouse (Figure 2). When we analysed fibroblasts from four familiar AD-type 1 (fAD 1) patients, we noted that the increase in the phosphorylation of APP Tyr residue was significantly greater than that in either healthy donors with a familiar history of fAD 1 or sporadic AD (Figure 3). Interestingly, the healthy donor #AG06846 did not show changes in APP Tyr phosphorylation levels when compared to the family member carrying the PSEN1 A246E genotype (Figure 4). However, due to the very small sample size of the single panel, the power of these statistical analyses appears to be very speculative. Of interest, Tyr phosphorylation levels were increased in AD and PSEN1 A246E patients when compared to patients affected by other neurodegenerative diseases (Figure 4).

In regard to the Fyn TK activity, fibroblasts in which APP Tyr phosphorylation levels were increased also showed elevated Fyn Tyr_420_ phosphorylation levels (Figure 1). However, Fyn Tyr_420_ residue was also phosphorylated in fibroblasts from PD patients (Figure 4) as well as in some healthy donors (Figure 1). The findings that Fyn activation is not restricted to AD patients appeared consistent with previous evidence, pointing to the involvement of Fyn in a multitude of processes related either to cellular functions or dysfunctions [13] and strengthening the importance to confine any potential treatments with Fyn TKI to those patients in which APP Tyr_682_ phosphorylation levels are elevated.

## 4. Discussion

Despite the increasing efforts to understand the pathophysiology of AD, there are many unresolved questions regarding how to diagnose the disease early and the treatments available to ameliorate the condition in affected patients. To date, no efficient therapy has been found, and the majority of the pharmacologic approaches are only palliative because none of them delay the pathology [14]. This lack of success is partly explained by the complex aetiology of AD which makes this pathology heterogeneous and unpredictable.

We previously showed the potential role of APP Tyr_682_ phosphorylation as a biomarker for APP dysregulation in trafficking and processing, and as a predictive factor for the increased production of neurotoxic Aβ peptide and neurodegenerative processes [2,3,5,6,7,8,9,10,13,15]. Indeed, the dysregulation of kinase activity has been associated with the progression of other neurodegenerative diseases, and has been proposed as a potential biomarker for other major proteinopathies besides AD, including PD [16,17,18] and Huntington’s disease [19,20].

Given the above scenario, in this study we explored the possibility of using peripheral cells, such as skin fibroblasts, to assess changes in APP Tyr_682_ phosphorylation levels. We found that APP Tyr_682_ phosphorylation increases in fibroblasts of AD patients. In addition, hyperactivated Fyn and elevated APP Tyr phosphorylation levels were detectable in the neurons from the same AD patients. 

It is well established that Fyn can play a dual role in both the Aβ and tau pathologies observed in AD [21], exacerbating Aβ neurotoxicity through cellular prion protein (PrP^C^) and interacting directly with tau [22] by phosphorylating it at Tyr_18_ [23]. We previously delineated a new possible mechanism in which Fyn initiated the amyloid cascade by phosphorylating the APP Tyr_682_ residue, which would then induce the amyloidogenic cleavage [2,13]. This cascade of events would amplify the neurotoxic effects of Aβ, tau hyperphosphorylation, neuronal tangles, and, consequently, neuronal damage. Consistently, we found here that Fyn was activated in #379 and #125 healthy donors and in PD patients (#446, #527), emphasising the involvement of Fyn in other processes besides AD [13]. This observation further strengthens the importance to restrict any potential treatments with Fyn TKI to those patients in which APP Tyr_682_ phosphorylation levels are elevated, thus preventing off-target effects, and improving the outcome of the disease.

Indeed, the identification of Fyn as an actor in APP Tyr_682_ phosphorylation in AD neurons opens the scenario to either test already commercially available Fyn TKIs or develop more specific compounds that can potentially control Fyn hyperactivity in AD patients and prevent Aβ_42_ production in patients in whom APP Tyr_682_ phosphorylation is increased.

Overall, the observations reported here prospect the possibility that events related to APP Tyr phosphorylation in fibroblasts may reflect Aβ related abnormalities in the brain, and that, if correlated with changes in cognitive performance and with other biomarkers including neuroimaging approaches or to the levels of Aβ and phosphorylated tau in biofluids, may allow the identification of patients who could benefit from a personalised pharmacologic approach, by using Fyn TKI. Indeed, these results underline the need to extend this evidence to a larger number of samples/fibroblasts, as well as to develop a procedure selective and quantitative for the detection of the levels of APP Tyr_682_ phosphorylation.

In conclusion, this study proposes to use changes in APP Tyr_682_ phosphorylation levels in peripheric cells, such as fibroblasts, as a potential diagnostic approach to design personalised therapy in patients in which APP Tyr_682_ phosphorylation is increased. Indeed, further studies will be necessary to extend this analysis to a larger number of patients.

## Figures and Tables

**Figure 1 brainsci-11-00103-f001:**
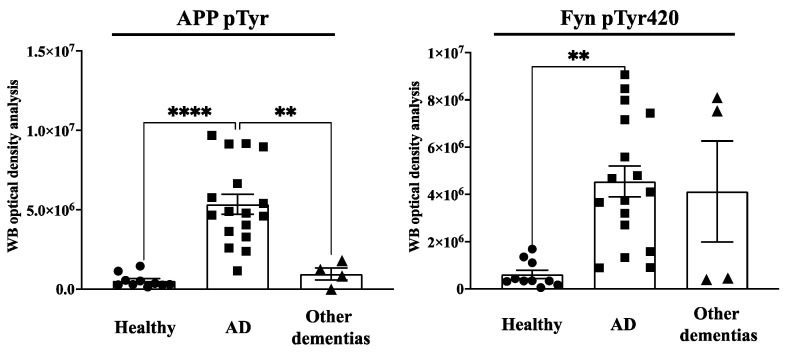
APP Tyr phosphorylation (APPpTyr) increases in fibroblasts from AD patients but not in patients with other forms of dementia or in healthy controls. The left panel reports the optical density analysis of APP pTyr bands, expressed as the mean optical density ratio between APP pTyr and APP basal levels from each sample in AD neurons (square) vs. healthy (circle) controls and other dementias (triangle). ** *p* < 0.005 and **** *p* < 0.0001, one-way ANOVA followed by Tukey’s test. The right panel reports the densitometric analysis of the bands expressed as the mean optical density (OD) ratio of Fyn pTyr_420_ relative to basal Fyn levels. ** *p* < 0.005, one-way ANOVA followed by Tukey’s test.

**Figure 2 brainsci-11-00103-f002:**
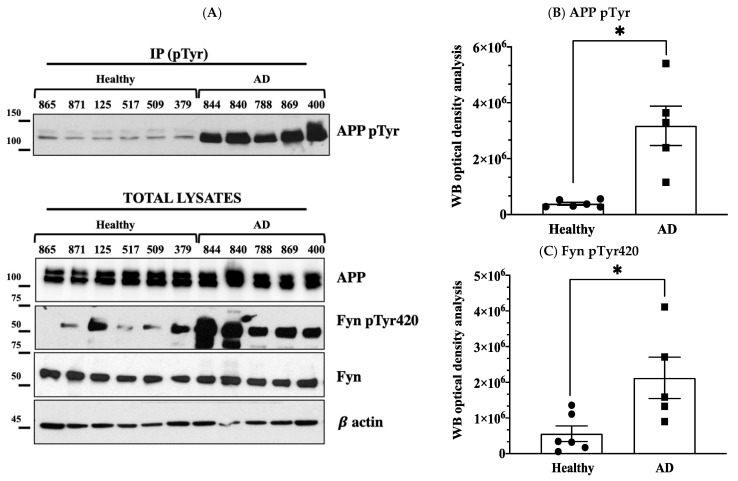
APP Tyr phosphorylation increases in fibroblasts from AD patients but not in healthy donors. Panel (**A**) reports data from six unaffected spouses (circle) and five AD cases (square). Panel (**B**) shows the optical density analysis of APP pTyr bands expressed as the mean optical density ratio between APP pTyr and APP basal levels from each sample in AD neurons vs. healthy controls. * *p* < 0.05; paired *t*-test. Panel (**C**) reports the ratio of pFyn Tyr_420_ relative to basal Fyn levels. * *p* < 0.05; unpaired *t*-test. Western blot (WB) APP and Fyn expression levels are reported in panels (**D**,**E**). β-actin values were used as loading controls.

**Figure 3 brainsci-11-00103-f003:**
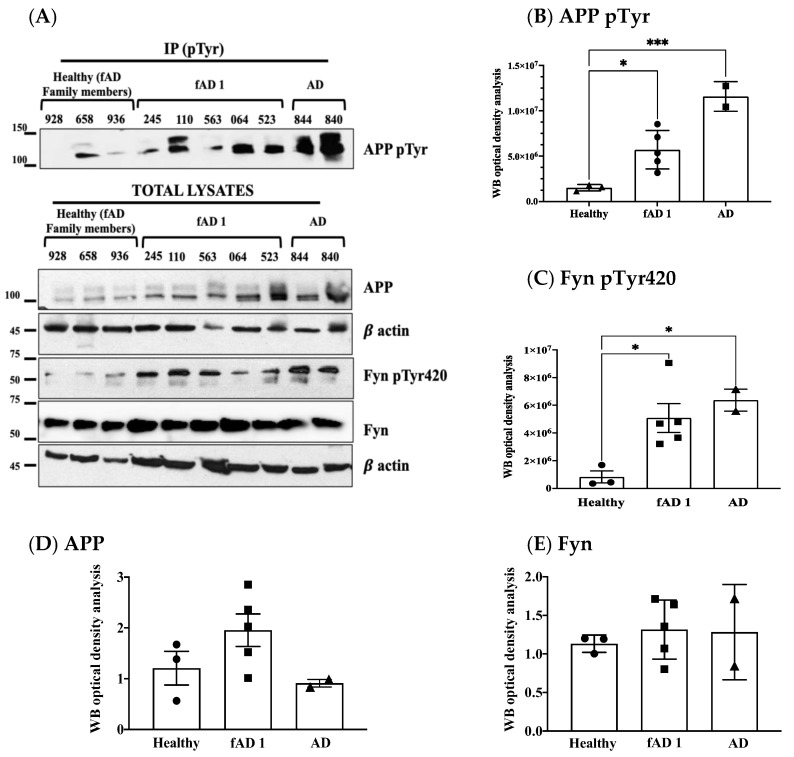
APP Tyr phosphorylation increases in fibroblasts from familial AD type 1 (fAD 1) and AD patients compared to healthy controls with a history of fAD 1. Panel (**A**) shows WB analyses of APP pTyr, Fyn pTyr420, APP and Fyn in healthy donors (circle) patients with a diagnosis of fAD 1 (square) and AD (triangle). Panels (**B**–**E**) report the densitometric analyses of the bands shown in panel (**A**). In particular, panel (**B**) reports the optical density analysis of APP pTyr bands expressed as the mean optical density ratio between APP pTyr and APP basal levels from each sample. * *p* < 0.05; *** *p* < 0.001. Panel (**C**) reports the ratio of FynpTyr420 relative to basal Fyn levels. * *p* < 0.05, one-way ANOVA followed by Tukey’s test. APP and Fyn expression levels, normalised on β-actin values, are reported in panels (**D**,**E**).

**Figure 4 brainsci-11-00103-f004:**
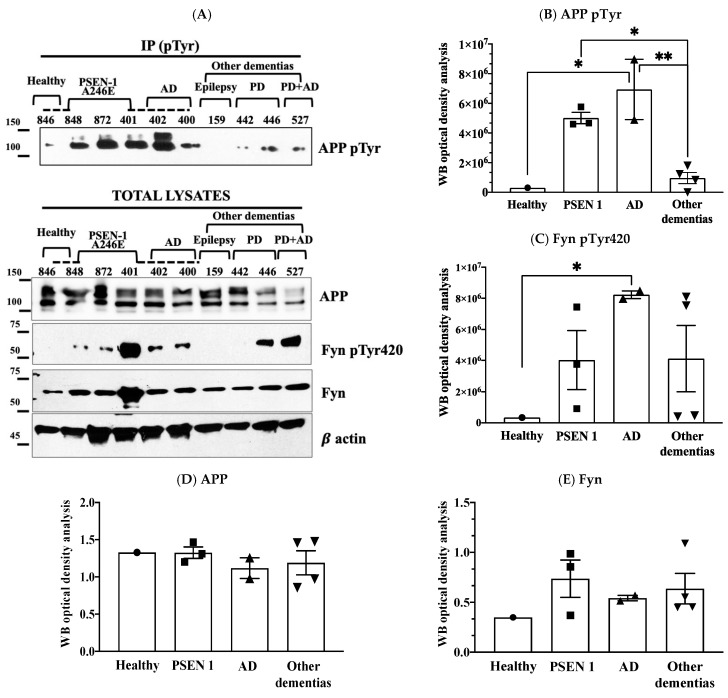
APP Tyr phosphorylation increases in fibroblasts from AD patients or in patients carrying PSEN 1 mutation, but not in patients with other forms of dementia or in healthy donors. Panel (**A**) shows WB analyses of APP pTyr, Fyn pTyr_420_, basal APP and Fyn levels. Panel (**A**) in particular reports data from one healthy donor #846 with his sibling #848 who carried PSEN1 A246E mutation (indicated with a discontinued line); the PSEN1 A246E genotype was also present in patients #872 and #401; patients #400, #401 and #402 were siblings, and are indicated by the discontinued line; patients #159 had a diagnosis of epilepsy for their entire life, however post-mortem histologic analysis revealed an AD phenotype; patients #442 and #446 were PD, whereas #527 was clinically diagnosed with PD, however after death brain histology indicated an AD phenotype. Panel (**B**) reports the optical density analysis of APP pTyr bands expressed as the mean optical density ratio between APP pTyr and APP basal levels from each sample in fAD 1 and AD fibroblasts vs. healthy controls and other dementias. * *p* < 0.005; ** *p* < 0.001, one-way ANOVA followed by Tukey’s test. Panel (**C**) reports the ratio of pFyn Tyr_420_ relative to basal Fyn levels. The graph represents the densitometric analysis of the bands expressed as the mean optical density (OD). * *p* < 0.005, one-way ANOVA followed by Tukey’s test. APP and Fyn expression levels are reported in panels (**D**,**E**). β actin values were used as loading controls.

**Table 1 brainsci-11-00103-t001:** Genotype, age, and sex of fibroblast donors. Individual clinical data (diagnostic criteria) and specific clinical features for the majority of donors are available on the Coriell website.

Healthy			
Genotype	Age (years)	Sex	Source (#)
Unaffected spouse	48	F	AG07865
	49	F	AG07871
	60	F	AG08379
	66	F	AG08517
	73	M	AG08509
	64	M	AG08125
fAD 1 Family members	69	F	AG07936
	47	F	AG07928
	42	M	AG08658
Family member with #AG06848	75	F	AG06846
**Alzheimer’s Patients**			
fAD 1	41	F	AG08110
	38	F	AG08563
	75	M	AG08245
	41	M	AG08064
	43	M	AG08523
PSEN 1 A246E	49	F	AG06840
	53	M	AG07872
	52	M	AG06848 family member of AG06846
AD	61	F	AG04400 sibling of AG04401and AG04402
	53	F	AG04401 sibling of AG04400 and AG04402
	60	F	AG06869
	49	M	AG06844
Apo E3/E4	87	F	AG10788
Apo E4/E4	47	M	AG04402 sibling of AG04400and AG04401
**Other Dementias**			
Parkinson’s	57	M	AG20446
	53	M	AG20442
Parkinson’s + AD	60	M	AG08527
Epilepsy + AD	52	F	AG04159

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
