# Peer review of "Might Fibroblasts from Patients with Alzheimer’s Disease Reflect the Brain Pathology? A Focus on the Increased Phosphorylation of Amyloid Precursor Protein Tyr682 Residue"

_brainsci, 2021, doi:10.3390/brainsci11010103_

Round 1

Reviewer 1 Report

Dear editor,

I have carefully read with great interest the study “Might fibroblasts from patients with Alzheimer’s disease reflect the brain pathology? Focus on the increased phosphorylation of Amyloid precursor protein Tyr682 residue.” by Iannuzzi et al. Overall, I find the work very interesting, the manuscript is very well-written with good and clear structure that allows the reader to easily follow the results. However, as the manuscript stands now I have some concerns on the experiments and especially at the statistical analysis that was performed, but I definitely believe that the authors should be given the chance to address these concerns and provide an improved revised version.

A general conceptual question is about the type of cells that are being used in the current study and the relevance to the disease-related pathology. As the authors nicely report, there is a good level of experimental data to support the importance of neuronal APP Tyr phosphorylation as an AD-related neurotoxic marker that precedes the disease phenotypes. However, the authors are using in the current study mature skin fibroblasts from patients where they detect the upregulation of this specific marker, so nothing related to the neurons where the pathology is abundant. As specific mutations or genetic inheritance (such as PSEN-1 mutants or the other fAD1 cases) could explain this observation in such irrelevant types of cells, how do the authors explain this phenotype in the fibroblasts from sporadic cases? Does this mean that the APP pTyr phenotype is not neuronal specific?

Below you can also find some more specific comments:

  • Line 62: what does it mean “TKI”? Is this Fyn tyrosine kinase inhibitors?
  • Table 1: I think it would be more helpful and easier for the reader to include lines separating the groups of cases or healthy controls. For example, it is not clearly obvious how many fAD1 patients were used or in other words at which number the fAD1 patients stop and the PSEN1 A246E patients begin. On the same note, there doesn’t seem to be a clear classification on the patients within the subgroups, perhaps clustering as male and female would make more sense.
  • Figure 1A: why is the patient with number ending 245 labelled separately from the other fAD1 patients? Also, in line 125 is mentioned that “increase in the phosphorylation of APP Tyr residue occurred in fibroblasts from all Familiar AD-type 1”, which is not absolutely accurate as the patient with number ending in 245 is an outlier.
  • Can the authors explain how they decided to compare the patients between them? The main reason behind my question is the potential variability between APP pTyr levels in healthy controls that might make the presented differences with the disease-derived ones higher or lower. For example, in panel A, if the authors decided to include only the healthy subject with number ending in 658, then obviously the fAD1 patients with numbers ending at 245, 110, and 563 would not have significantly higher levels of APP pTyr. To be more specific:
    • Figure 1A: I think it makes sense to compare the fAD1 cases with their healthy family members, because this is a direct comparison. However, since the authors decide to include sporadic AD cases, I think it would make sense to include also healthy unrelated controls.
    • Figure 1B: I think this panel is very interesting because the reader can see the direct blood relationships between patients as they are nicely depicted with the dotted line. However, in the previous panel there are 3 healthy controls and in the next panel (C) there 6. Therefore, I think having only 1 healthy subject in a western blot with 9 diseased makes it a slightly unfair comparison.
  • Figure 1D and E: the statistical analysis is my major concern for the current study. I would like to point out that with this concern I am not challenging the finding of the authors, as there seem to be clear differences in the APP pTyr levels, I am just not very confident with the way their statistical analysis was done. In principle, the authors need to explain how they are able to compare the different groups since they are not in the same western blot analysis. In my point of view, the currently presented statistical analysis is not valid unless:
    • unless the authors are able to have an absolute quantification such as the protein concentration or
    • have blotted all the cases together in the same western blots

More specifically, based on the presented western blots only the following direct comparisons can be made for statistical purposes:

Panel A: Healthy 928, 658, 936 compared with the diseased 245, 110, 563, 064, 523, 844, 840

Panel B: only 1 healthy subject (846) with 3 PSEN-1 mutation-carriers (848, 872, 401), 2 AD (402, 400) and 4 labelled as “other dementias” (159, 442, 446, 527)

Panel C: 6 healthy controls (865, 871, 125, 517, 509, 379) with 5 AD patients (844, 840, 788, 869, 400).

Author Response

I have carefully read with great interest the study “Might fibroblasts from patients with Alzheimer’s disease reflect the brain pathology? Focus on the increased phosphorylation of Amyloid precursor protein Tyr682 residue.” by Iannuzzi et al. Overall, I find the work very interesting, the manuscript is very well-written with good and clear structure that allows the reader to easily follow the results. However, as the manuscript stands now I have some concerns on the experiments and especially at the statistical analysis that was performed, but I definitely believe that the authors should be given the chance to address these concerns and provide an improved revised version. 

A general conceptual question is about the type of cells that are being used in the current study and the relevance to the disease-related pathology. As the authors nicely report, there is a good level of experimental data to support the importance of neuronal APP Tyr phosphorylation as an AD-related neurotoxic marker that precedes the disease phenotypes. However, the authors are using in the current study mature skin fibroblasts from patients where they detect the upregulation of this specific marker, so nothing related to the neurons where the pathology is abundant. As specific mutations or genetic inheritance (such as PSEN-1 mutants or the other fAD1 cases) could explain this observation in such irrelevant types of cells, how do the authors explain this phenotype in the fibroblasts from sporadic cases? Does this mean that the APP pTyr phenotype is not neuronal specific? 

First, I would like to thank this reviewer for all the crucial aspects he/she pointed out.  

We completely agree about most of his/her concerns. This is the reason why we decided to present these results as perspective/commentary rather than in other forms of publication and I included in the title “might” to underline the need to further study most of the aspects that we are here showing. Indeed, I wouldn’t say that APP pTyr phenotype is not neuronal specific, as APP is expressed in all the cell types, but rather that we are opening a new scenario in which changes in APP Tyr phosphorylation in peripheral cells may become “relevant” in predicting, anticipating or recapitulating events in brain such as Ab production. Of course, at this point these assumptions are mere speculation.  

Below you can also find some more specific comments: 

Line 62: what does it mean “TKI”? Is this Fyn tyrosine kinase inhibitors? 

TKI refers to Tyrosine kinase inhibitors as reported in line 52 

Table 1: I think it would be more helpful and easier for the reader to include lines separating the groups of cases or healthy controls. For example, it is not clearly obvious how many fAD1 patients were used or in other words at which number the fAD1 patients stop and the PSEN1 A246E patients begin.  

On the same note, there doesn’t seem to be a clear classification on the patients within the subgroups, perhaps clustering as male and female would make more sense. 

We followed Reviewer suggestion and included lines to separate the different groups reported in the Table, although this might not follow the journal editing style.  

Male and female have been listed separately in this new version. 

Figure 1A: why is the patient with number ending 245 labelled separately from the other fAD1 patients? Also, in line 125 is mentioned that “increase in the phosphorylation of APP Tyr residue occurred in fibroblasts from all Familiar AD-type 1”, which is not absolutely accurate as the patient with number ending in 245 is an outlier. 

We apologize for this inaccuracy in reporting data regarding patient #245. In this new version we amended this mistake and consequently statistical analysis has been corrected. 

Can the authors explain how they decided to compare the patients between them? The main reason behind my question is the potential variability between APP pTyr levels in healthy controls that might make the presented differences with the disease-derived ones higher or lower. For example, in panel A, if the authors decided to include only the healthy subject with number ending in 658, then obviously the fAD1 patients with numbers ending at 245, 110, and 563 would not have significantly higher levels of APP pTyr. To be more specific: 

Figure 1A: I think it makes sense to compare the fAD1 cases with their healthy family members, because this is a direct comparison. However, since the authors decide to include sporadic AD cases, I think it would make sense to include also healthy unrelated controls. 

Figure 1B: I think this panel is very interesting because the reader can see the direct blood relationships between patients as they are nicely depicted with the dotted line. However, in the previous panel there are 3 healthy controls and in the next panel (C) there 6. Therefore, I think having only 1 healthy subject in a western blot with 9 diseased makes it a slightly unfair comparison. 

Figure 1D and E: the statistical analysis is my major concern for the current study. I would like to point out that with this concern I am not challenging the finding of the authors, as there seem to be clear differences in the APP pTyr levels, I am just not very confident with the way their statistical analysis was done. In principle, the authors need to explain how they are able to compare the different groups since they are not in the same western blot analysis. In my point of view, the currently presented statistical analysis is not valid unless: 

unless the authors are able to have an absolute quantification such as the protein concentration or 

have blotted all the cases together in the same western blots 

More specifically, based on the presented western blots only the following direct comparisons can be made for statistical purposes: 

Panel A: Healthy 928, 658, 936 compared with the diseased 245, 110, 563, 064, 523, 844, 840 

Panel B: only 1 healthy subject (846) with 3 PSEN-1 mutation-carriers (848, 872, 401), 2 AD (402, 400) and 4 labelled as “other dementias” (159, 442, 446, 527) 

Panel C: 6 healthy controls (865, 871, 125, 517, 509, 379) with 5 AD patients (844, 840, 788, 869, 400). 

We truly appreciated these criticisms and we used reviewer suggestions to improve the way of presenting our data.  

In this revised version we expressed data as absolute values instead of % of CTRL. This should help to combine results from different WB (Figure 1). Indeed, combining WB from different experiments is essential due to the difficulties to load so many samples on the same gel. Additionally, it wouldn’t have any potential clinical relevance the single gel analysis 

However, we agree with reviewer comments regarding the importance to include specific patient information to better interpret our resultsThis is a very critical aspect that we acknowledged when we tried to incorporate family and genetic background in our IP/WB analysis, and to interpret our findings in respect to that information, that is probably the reason why so many clinical trials failed in addressing their endpoint. However, this is again only speculation. 

In order to accomplish the reviewer concerns, in this revised version we also included quantification for each single WB. As the reviewer expected some results did not show statistical significance because of the limited sample size, however, I believe they keep the overall message that changes in APP Tyr682 phosphorylation can represent a potential peripheral cell biomarker of AD 

Reviewer 2 Report

To the Authors:

In the present manuscript, Iannuzzi and colleagues use fibroblasts as a framework to investigate APP Tyr682 phosphorylation in a cohort of AD patients. Using this cellular model, the authors report that APP Tyr682 phosphorylation levels are consistently higher in AD patients compared to fibroblasts from healthy controls or patients from other neurodegenerative disorders such as Parkinson's disease (PD). Based on their results, the authors claim that APP Tyr682 hyperphosphorylation can be used as a potential diagnostic biomarker for AD.

Overall, the manuscript presents interesting data and might be relevant for AD biomarker research.

There are, however, a few issues that must be addressed by authors, as described below.

  • Could the authors provide more information about the PD patients? Were they familial or idiopathic PD patients? If familial PD, which mutations?
  • Dysregulation of kinase activity/neuronal protein hyperphosphorylation has been associated with disease progression, and have been proposed as potential biomarkers for other major proteinopathies besides AD, including PD (PMID: 29961428; PMID: 31881263; PMID: 28648742) and Huntington’s disease (PMID: 20460152; PMID: 25941073). The authors should acknowledge these studies in their manuscript, and put them into perspective with their own hypothesis.
  • Western blots in Fig. 1 are overexposed; protein bands are completely black, indicating potential saturation of the chemiluminescent signal. Since the exposure time and the linearity in chemiluminescent immunodetection in western blotting is critical, the authors should provide less developed western blots.

Author Response

In the present manuscript, Iannuzzi and colleagues use fibroblasts as a framework to investigate APP Tyr682 phosphorylation in a cohort of AD patients. Using this cellular model, the authors report that APP Tyr682 phosphorylation levels are consistently higher in AD patients compared to fibroblasts from healthy controls or patients from other neurodegenerative disorders such as Parkinson's disease (PD). Based on their results, the authors claim that APP Tyr682 hyperphosphorylation can be used as a potential diagnostic biomarker for AD.Overall, the manuscript presents interesting data and might be relevant for AD biomarker research. 

There are, however, a few issues that must be addressed by authors, as described below. 

  • Could the authors provide more information about the PD patients? Were they familial or idiopathic PD patients? If familial PD, which mutations? 

We thank the reviewer for his/her comments and suggestions. In this revised version, we included in the Method section a few more information about the three PD patients. According to the information available on Coriell website, all the PD patients included in the study were sporadic. 

  • Dysregulation of kinase activity/neuronal protein hyperphosphorylation has been associated with disease progression, and have been proposed as potential biomarkers for other major proteinopathies besides AD, including PD (PMID: 29961428; PMID: 31881263; PMID: 28648742) and Huntington’s disease (PMID: 20460152; PMID: 25941073). The authors should acknowledge these studies in their manuscript, and put them into perspective with their own hypothesis. 

A sentence mentioning these articles have been included in the discussion. 

  • Western blots in Fig. 1 are overexposed; protein bands are completely black, indicating potential saturation of the chemiluminescent signal. Since the exposure time and the linearity in chemiluminescent immunodetection in western blotting is critical, the authors should provide less developed western blots. 

APP WB have been replaced, accordingl

Round 2

Reviewer 1 Report

Dear editor,

I have carefully read the revised version of the manuscript by Iannuzzi et al and overall, I am very happy with the modifications. The reviewers addressed elegantly all my concerns and from my side I do not see any reason why the study should not proceed with publication.

Reviewer 2 Report

The authors have successfully addressed all my comments, and I therefore fully support the present manuscript for publication in Brain Sciences.